# A Quasi-Experimental Pre-Post Assessment of Hand Hygiene Practices and Hand Dirtiness Following a School-Based Educational Campaign

**DOI:** 10.3390/ijerph22081198

**Published:** 2025-07-31

**Authors:** Michelle M. Pieters, Natalie Fahsen, Christiana Hug, Kanako Ishida, Celia Cordon-Rosales, Matthew J. Lozier

**Affiliations:** 1Center for Health Studies, Universidad del Valle de Guatemala, Guatemala City 01015, Guatemalaccordon@uvg.edu.gt (C.C.-R.); 2Division of Foodborne, Waterborne, and Environmental Diseases, National Center for Emerging and Zoonotic Infectious Diseases, Centers for Disease Control and Prevention (CDC), Atlanta, GA 30329, USA; cmariehug@gmail.com (C.H.); kanakoi@g.ucla.edu (K.I.); wfu2@cdc.gov (M.J.L.); 3Oak Ridge Institute for Science and Education (ORISE), Oak Ridge, TN 37830, USA; 4United States Public Health Service, Silver Springs, MD 20910, USA

**Keywords:** hand hygiene, COVID-19, education campaign, behavior change, hand dirtiness

## Abstract

Hand hygiene (HH) is essential for preventing disease transmission, particularly in schools where children are in close contact with other children. This study evaluated a school-based intervention on observed HH practices and hand cleanliness in six primary schools in Guatemala. Hand cleanliness was measured using the Quantitative Personal Hygiene Assessment Tool. The intervention included (1) HH behavior change promotion through Handwashing Festivals, and (2) increased access to HH materials at HH stations. Handwashing Festivals were day-long events featuring creative student presentations on HH topics. Schools were provided with soap and alcohol-based hand rub throughout the project to support HH practices. Appropriate HH practices declined from 51.2% pre-intervention to 33.1% post-intervention, despite an improvement in median Quantitative Personal Hygiene Assessment Tool scores from 6 to 8, indicating cleaner hands. Logistic regression showed higher odds of proper HH when an assistant was present. The decline in HH adherence was likely influenced by fewer assistants and changes in COVID-19 policies, while improvements in hand cleanliness may reflect observational bias. These findings emphasize the importance of sustained behavior change strategies, reliable HH material access, and targeted interventions to address gaps in HH practices, guiding school health policy and resource allocation.

## 1. Introduction

Appropriate hand hygiene practices, such as washing hands with soap and water or applying alcohol-based hand rub (ABHR) have the potential to reduce diarrhea and respiratory infections among school-aged children [1]. These infections impact child health and increase absenteeism, which has a negative impact on children’s educational outcomes [2]. Despite the recognized importance of hand hygiene (HH), many settings, especially in low- and middle-income countries (LMICs), still lack the necessary infrastructure, like access to water and handwashing facilities. According to a report from 2021 by the World Health Organization (WHO) and UNICEF on HH, 462 million children attended schools with no HH facilities [3].

Besides having access to appropriate resources, one of the most efficient ways to encourage HH practices in children is through school-based activities, such as demonstrations, games, and songs [4]. For decades the WHO has emphasized the role that schools have in promoting healthy habits, such as HH, because they play a key role in facilitating behavior change by teaching children why HH is important and providing opportunities for them to practice these skills [5]. Additionally, childhood is an important time to target HH practices, as habits are not yet firmly set [6].

Several school-based HH interventions in LMICs have demonstrated increased HH practices among students through a combination of infrastructure improvements and interactive educational activities. A study in Kenya demonstrated that a school-based HH intervention, which included curriculum delivery, resulted in sustained improvements in students’ hygiene practices, increased knowledge, and reduced respiratory infections [7]. Similarly, a study in Uganda demonstrated that a skill-based HH education was more effective at influencing HH compliance among students than traditional HH education [8].

While such interventions have shown promising results internationally, there is limited evidence on school-based HH interventions in Guatemala. According to the Joint Monitoring Programme for Water Supply, Sanitation, and Hygiene, in 2021 Guatemala did not have sufficient data to estimate the availability of basic hygiene services in schools. However, previous studies in the country have shown that primary school students generally demonstrate high HH knowledge and positive attitudes, though they fail to correctly identify key moments for HH, such as after restroom use or after coughing or sneezing [9,10].

Studying HH practices and interventions in Guatemala is particularly important given the country’s high burden of preventable infectious diseases, deep rural-urban and socioeconomic disparities in access to WASH services, and the large proportion of school-aged children [11,12]. Evidence from Guatemala can inform context-specific solutions and contribute to broader regional and global efforts to improve school-based hygiene and health outcomes. In this context, we implemented a school-based HH intervention that combined increased access to HH materials with behavior change activities. This evaluation had two main objectives: first, to provide updated evidence on HH practices among Guatemalan schoolchildren using data from six public primary schools; and second, to assess the effectiveness of the intervention in improving HH practices and reducing hand dirtiness.

## 2. Materials and Methods

### 2.1. Study Setting

This study employed a quasi-experimental, pre-post design to assess changes in HH practices among students aged 8 to 13 years before and after an intervention. Six public primary schools were selected by convenience based on Universidad del Valle de Guatemala’s (UVG) previous presence in the department of Quetzaltenango, Guatemala due to their epidemiologic surveillance program at the local hospitals and according to selection criteria (being open with in-person classes according to the Ministry of Education’s COVID-19 epidemiological surveillance and willingness to participate in the study). Two schools were classified as urban, and four schools were classified as rural. The intervention was implemented from October 2022 to April 2023 at six primary schools in three different municipalities (San Miguel Sigüilá, San Juan Ostuncalco, and Concepción Chiquirichapa) of Quetzaltenango (Figure 1). Before and after the intervention, two types of data were collected at the schools to benchmark their HH practices using HH observations and hand dirtiness evaluations. Pre-intervention data were collected during May–June 2022, and post-intervention data were collected in July 2023.

### 2.2. Intervention

The intervention included two components: (1) HH behavior change promotion activities, called Handwashing Festivals and (2) increased access to HH materials at HH stations. The goal of the intervention was to increase the rate at which students perform appropriate HH. The HH behavior change intervention included activities that focused on teaching students about the importance of HH practice and the critical moments for this practice, through handwashing festivals. These festivals were day-long community events to promote HH, organized and hosted by each of the six participating schools. Each grade or class was tasked with presenting a HH topic, such as critical times for handwashing, handwashing steps, and/or germ transmission. Schools had the freedom to choose how they presented their assigned HH topics. This could include posters, dances, songs, puppet shows, or any other creative means to convey the information effectively. Our team provided support by supervising activities to ensure that they were conducted in accordance with the project’s objectives. The festivals encouraged interactive learning, allowing students to actively participate in presentations and activities. Handwashing festivals were carried out twice at each school between pre- and post-intervention assessments, once in October 2022 and again in April 2023. To ensure consistent access to the HH materials, the intervention also provided each school with soap and ABHR to facilitate HH practices during the entire duration of the project.

### 2.3. Procedures

#### 2.3.1. Hand Hygiene Observations Methods

We observed HH practices of students at three different key-times during the school day: as they entered school, after restroom use, and before eating. We selected these timepoints due to public health requirements set by the Ministry of Health and Ministry of Education during COVID-19 pandemic as part of their mitigation efforts. To conduct the observations, enumerators stood near the school entrances, outside the bathrooms, and around the classrooms, respectively. HH at school entrances was obligatory per school and government guidance, thus it was selected as the reference group for analyses. Students were aware that a team was observing them but were not told what was being observed. During each period, students were observed until 20 observations were carried out, or for a predetermined amount of time (i.e., the duration of recess), whichever came first. Enumerators documented (1) whether students attempted HH; (2) the type of HH attempted (handwashing with soap and water, handwashing with only water, or using ABHR); (3) the handwashing duration (≥20 s or <20 s); (4) the student’s sex; (5) the presence of an HH attendant; (6) and the HH materials available. Observations were only carried out if appropriate HH materials were available (soap and water, and/or ABHR). Data were collected on paper, and later entered into REDCap [13,14]. We determined a sample size based on the duration of the observation period (i.e., however long recess lasted) or a maximum of 20 students observed per period. Therefore, we targeted a sample size of 360 observations (60 observations per school—20 observations maximum per observation period) at each pre- and post-intervention.

#### 2.3.2. Hand Dirtiness Evaluation Methods

To assess students’ hand dirtiness, we used the Quantitative Personal Hygiene Assessment Tool (qPHAT) [15]. The methodology involves tracing the palm and fingertips of the participant’s hand with a pre-moistened Hygea (PDI, Orangeburg, NY, USA) sterile saline gauze pad and comparing the darkest half-square inch area of the gauze pad to the qPHAT color scale [15] (Figure 2). The qPHAT uses a scale from 0–10 where 0 indicates the most visible dirt possible and 10 indicates the absence of any visible debris or dirt. Hand swabs were collected at three different times during the day: as students entered school, after restroom use, and before eating. Students were selected by convenience as they were walking into school, going to the restroom, or at recess. After the sample was collected, the score on the qPHAT scale was determined and agreed upon by two enumerators. If the value fell between two numbers, the higher value was recorded, as per the qPHAT methodology instructions. Scores were entered into REDCap along with students’ demographic information (age, grade, and sex), and an image of the swab. Parental consent and students’ assent was obtained before taking the swabs. To detect a significant difference in hand dirtiness pre- and post-intervention with 95% confidence, we estimated that we needed to collect 21 hand swabs per school at pre-and post-intervention for a total of 126 swabs pre-intervention and 126 swabs post-intervention.

### 2.4. Data Analysis

Appropriate HH practice was calculated based on whether observed students washed hands with water and soap for at least 20 s or used ABHR to clean their hands at specific times. We disaggregated appropriate HH practice by key characteristics of the students and HH opportunities (time of observation, sex, HH attendant, and school location) and conducted chi-square tests to test statistical significance in the differences in practices between pre- and post-intervention. We compared the mean hand dirtiness (qPHAT) scores between pre- and post-intervention using independent *t*-tests, as the observations at the two time points involved different groups of students rather than paired measurements. We merged baseline and endpoint data and used logistic regression to assess factors associated with appropriate hand hygiene (HH) performance (dependent variable). Independent variables included sex, HH moment, locality, and evaluation time point (pre-intervention or post-intervention). All analyses were conducted using STATA version 17 (StataCorp LLC; College Station, TX, USA). A *p*-value of <0.05 was considered statistically significant for all analyses.

### 2.5. Ethics Statement

All study activities were in accordance with the ethical standards of the institutional review board of the Center for Health Studies of the Universidad del Valle de Guatemala (UVG). The study protocol was reviewed and approved by UVG’s ethical committee on 5 May 2021 (No. 246-05-2021). This activity was reviewed by the U.S. Centers for Disease Control and Prevention (CDC), deemed research not involving human subjects, and was conducted consistent with applicable federal law and CDC policy. Data was collected from 7 June 2021, to 17 November 2023. Written informed consent was obtained from parents to explicitly indicate “yes” or “no” regarding their child’s participation and were returned to school. Only students whose parents/legal guardians provided written informed consent were included in the study. Verbal assent was obtained from students participating in hand swabs.

## 3. Results

### 3.1. Hand Hygiene Observations

Pre-intervention, five of the six participating schools had a HH station with ABHR available at entrances for student use. At all schools, water and soap were available for students to practice HH before eating and after restroom use. Post intervention, only two schools had HH stations available at school entrances—one school provided ABHR, while another provided soap and water. For observations carried out before eating and after restroom use, all schools had soap and water available.

We conducted 289 HH observations pre-intervention and 275 observations post-intervention (Table 1). At baseline, 52.2% (n = 151) of students observed were male. Most (42.9% n = 124) of the observations were carried out before eating, followed by after restroom use (29.1%, n = 84) and at school entrances (28.0%, n = 81). During post-intervention, 58.2% (n = 160) of students observed were male. Only 12.7% (n = 35) of observations were done at school entrances, followed by 43.6% (n = 120) after restroom use and 43.6% (n = 120) before students ate. At pre-intervention a HH assistant was present during 44.6% (n = 129) of the observations, compared to only 10.0% (n = 27) post-intervention. Most observations at pre- and post-intervention were carried out at rural schools (67.8% and 63.6%, respectively).

Despite the availability of water and soap or ABHR at both time points, 51.2% (n = 148) of students performed appropriate HH pre-intervention, which significantly declined to 33.1% (n = 91) of students’ post-intervention (Table 2). We also observed significant differences in appropriate HH practices between the pre- and post-intervention groups across all covariates.

We ran a logistic regression model to explore relationships between HH time, sex, HH assistance, rural versus urban, and appropriate HH performance (See Table 3). The odds of performing appropriate HH were lower after using the restroom (aOR 0.41; 95% CI: 0.20–0.80) and before eating (aOR 0.18; 95% CI: 0.09–0.34) compared to upon entering the school. The odds of performing appropriate HH was much higher when an attendant was present (aOR 6.69; 95% CI 3.74–11.94). The odds of performing appropriate HH was lower in urban areas than rural areas (aOR 0.42; 95% CI 0.27–0.65). There was no significant difference in performance of appropriate HH between girls and boys (aOR 0.76; 95% CI 0.51–1.13) or between baseline and endpoint (aOR 0.95; 95% CI 0.62–1.45). We tested for interaction between the intervention and the other co-variates, but for each interaction, we either did not find any significant interaction, or there were not enough observations to accurately assess interaction.

### 3.2. Hand Dirtiness Evaluation

Pre-intervention, we collected 82 swabs, of which 50% (n = 41) were collected from male students. The majority of the swabs were collected from students aged 9–12 years (86.6%). Swabs were collected in the following distributions: 35.5% (n = 29) as students were entering the school, 31.7% (n = 26) after restroom use, and 32.9% (n = 27) before eating. In terms of school location, 72% (n = 59) of the swabs were collected at rural schools, and 28% (n = 23) at urban schools (Table 4).

Post-intervention, we collected 126 swabs, of which 61.9% (n = 78) were collected from male students. Most swabs were collected from students in the 9–12 age range (69.8%). Forty-two samples were collected at each time: school entrances, after using restroom, and before eating. 66.7% (n = 84) of samples were collected at rural schools, and 33.3% (n = 42) of samples were collected at urban schools. The median qPHAT score increased from six pre-intervention to eight post-intervention (Table 5). The qPHAT scores were also higher at post-intervention than pre-intervention among male and female students, at the entrance, after restroom use, and among students in urban schools.

In a logistic regression model measuring the odds of scoring greater than the overall median (7), students were more likely to score higher than the median after using the restroom compared to at entrances to schools (aOR: 11.20; 95% CI: 4.63–27.09). Boys were more likely to score greater than the median than girls (aOR 2.04; 95% CI: 1.06–3.93). Lastly, students at the endpoint were more likely to score greater than the median than students at baseline (aOR 2.66; 95% CI: 1.35–5.21). 

## 4. Discussion

This assessment evaluated the change in appropriate HH practices among students in six schools in Guatemala before and after an intervention to increase access to HH materials and HH practices. Results from the logistic regression model showed no significant change in observed HH practices from pre to post intervention. Additionally, we saw a decrease in hand dirtiness at post evaluation according to qPHAT data. It is important to consider the timing of pre- and post-assessments in relation to the COVID-19 pandemic in understanding these results. Pre-intervention data were collected in 2022, when COVID-19 prevention messaging was prevalent and consistent. Studies have shown that constant exposure to health messaging can have an impact on knowledge [16], which may, in turn, influence practice. Many COVID-19 messages in Guatemala emphasized the harmful physical and social consequences of not complying with certain recommendations such as handwashing and mask wearing, employing a fear appeal that can act as a potential influencer of behavior [17,18,19]. When pre-intervention data was collected in June 2022, there were 53,405 active COVID-19 cases reported that month. Post-intervention data were collected in July 2023, after the WHO had declared an end to the COVID-19 global health emergency [20]. At this time, 5484 active cases were reported for July [21]. Guatemalan COVID-19 regulations were also lifted this month and no longer required HH stations or HH assistants to be available at school entrances or for schools to provide ABHR to students [22]. Due to these regulations, during the pre-intervention phase, HH assistants were predominantly stationed at school entrances. This placement likely contributed to the observed increase in HH adherence at this location compared to other areas (restrooms, and recess areas), where HH assistants were less frequently present. However, at post-intervention phase, when HH was no longer mandatory, the number of HH assistants was reduced. The shift in the distribution of HH assistants likely accounts for the observed decrease in overall HH adherence in unadjusted analyses. However, when adjusting for location and the presence of HH assistants in the logistic regression model, this association was diminished, suggesting that both the presence of assistants and the entrance-specific HH policy were significant factors in promoting HH compliance.

The qPHAT showed that students had less dirt on their hands post-intervention than they did pre-intervention. We acknowledge that bias might have an influence on these results. When carrying out the HH observations, students were aware that they were being observed but did not know what was being observed, whereas when collecting the hand swabs, students recognized the study team and data collection method, which could have resulted in students’ washing their hands more thoroughly before having their hands swabbed. Data on whether students had or had not washed their hands before the swabs were not considered, recorded, or selected for, which limits our ability to assess these patterns. Consequently, it is difficult to understand how much the observed improvement in hand dirtiness from pre to post can be attributed to the intervention.

The Handwashing Festivals can be a useful tool for developing educational campaigns to improve HH practices. The festivals were well-received by the schools, as evidenced by the amount of effort each school poured into their planning and execution. Evaluating the impact of these types of interventions is important to adjust and re-evaluate the components that can impact behavior change. Evidence suggests that information is better retained when health communication includes visuals [23]. Even though the Handwashing Festivals included visual activities, such as posters, dances, murals, and performances, the knowledge generated through these activities might not have been sufficient for behavior change [24]. Studies have shown that effective behavior change requires more than just education and informational campaigns, as perceived health benefits and knowledge are not strong enough determinants of handwashing behavior change [24,25,26,27]. Future interventions should include not only a consistent supply of HH materials, training, and education of teachers at schools but also include other behavior incentives such as environmental nudges [28]. Nudges specifically designed to promoting HH practices in schools include footpaths from toilets to handwashing stations or placing colorful signs above handwashing stations [29].

This study has several important limitations. First, the use of convenience sampling to select students for both observations and swab collection may have introduced selection bias. Second, because schools were selected based on accessibility and existing partnerships rather than random sampling, the findings may not be generalizable to other regions of Guatemala. Additionally, the quasi-experimental design without a control group means we could not account for secular trends, such as changing perceptions of COVID-19 risk over time, which likely influenced the observed reduction in appropriate HH practices. Moreover, allowing schools to choose how they presented their assigned HH topics may have introduced variability in implementation and potentially affected outcomes, although this approach was intended to foster ownership and sustainability

Lastly, the study was conducted during the COVID-19 pandemic, a period marked by specific regulations and heightened public health awareness, including mandatory HH stations and monitoring. The lifting of these regulations, including the removal of mandatory HH at entrances and assistants, may have impacted students’ willingness to perform hygiene behaviors voluntarily. This introduces a significant potential bias, and findings may not be fully extrapolatable to non-pandemic conditions. However, this also highlights the influence of government policies and regulatory environments on health behaviors, an important factor for interpreting results and planning future interventions.

## 5. Conclusions

HH practices seem to be mostly influenced by the presence of a HH assistant, and COVID-19 governmental measures. Policy is influential in HH practices because it removes individual choice and ensures consistency because it can be monitored and enforced. Future efforts should prioritize developing and testing targeted interventions that address both behavioral and structural factors influencing HH, such as supporting HH assistants, improving access to water and sanitation facilities, and strengthening policy enforcement. Collaborations between schools, families, local health authorities, and policymakers will be crucial to designing sustainable and culturally appropriate programs. Additionally, longitudinal and randomized studies are needed to evaluate the long-term effectiveness and scalability of this intervention, especially beyond pandemic contexts. By doing this, stakeholders can better allocate resources and implement strategies that not only improve HH practices but also contribute to reducing infectious diseases and enhancing child health and educational outcomes across Guatemala and similar settings.

## Figures and Tables

**Figure 1 ijerph-22-01198-f001:**
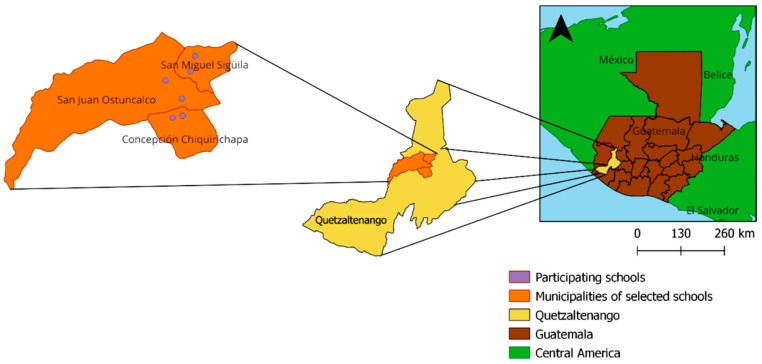
Map of evaluation school locations.

**Figure 2 ijerph-22-01198-f002:**

11-point qPHAT color scale.

**Table 1 ijerph-22-01198-t001:** Characteristics of hand hygiene observations.

Characteristic	Pre-Intervention n (%) N = 289	Post-Intervention n (%) N = 275
Sex		
Male	151 (52.3)	160 (58.2)
Female	138 (47.8)	115 (41.8)
Time		
School entrance	81 (28.0)	35 (12.7)
After restroom	84 (29.1)	120 (43.6)
Before eating	124 (42.9)	120 (43.6)
HH assistant present		
Yes	129 (44.6)	27 (10.0)
No	160 (55.4)	242 (89.9) *
School location		
Urban	93 (32.2)	100 (36.4)
Rural	196 (67.8)	175 (63.6)

* 6 observations were missing data, n = 269.

**Table 2 ijerph-22-01198-t002:** Comparison of appropriate hand hygiene practices pre- and post-intervention.

Covariate	Appropriate HH Practices (Pre-Intervention) n (% *)	Appropriate HH Practices (Post-Intervention) n (% *)	Percent Change	*p*-Value
Appropriate HH practices	148 (51.2)	91 (33.1)	−18.1	*p* < 0.001
HH Time				
At entrance	80 (98.8)	15 (42.9)	−55.9	*p* < 0.001
After restroom	36 (42.9)	30 (25.0)	−17.9	*p* < 0.01
Before eating	32 (25.8)	46 (38.3)	12.5	*p* < 0.05
Sex				
Male	67 (44.4)	46 (28.8)	−15.6	*p* < 0.01
Female	81 (58.7)	45 (39.1)	−19.6	*p* < 0.001
HH assistant				
Yes	97 (75.2)	27 (100.0)	24.8	*p* < 0.05
No	51 (31.9)	64 (26.5)	−5.5	*p* < 0.01
School location				
Urban	34 (36.6)	27 (27.0)	−9.6	*p* < 0.001
Rural	114 (58.2)	64 (36.6)	−21.6	*p* < 0.001

* Denominator is the total number of observations where materials to perform. Appropriate HH were available for each category (see Table 1).

**Table 3 ijerph-22-01198-t003:** Odds ratios for performing appropriate hand hygiene.

	OR (95% CI)	*p*-Value	aOR (95% CI)	*p*-Value
HH Time				
At Entrance *	Ref.		Ref.	
After Restroom	0.15 (0.09–0.26)	<0.0001	0.41 (0.20–0.80)	0.0099
Before Eating	0.08 (0.04–0.13)	<0.0001	0.18 (0.09–0.34)	<0.0001
Sex				
Female	Ref.		Ref.	
Male	0.58 (0.41–0.81)	0.0013	0.76 (0.51–1.13)	0.174
HH Assistance				
No	Ref.		Ref.	
Yes	9.67 (6.20–15.09)	<0.0001	6.69 (3.74–11.94)	<0.0001
Locality				
Rural	Ref.		Ref.	
Urban	0.51 (0.35–0.72)	0.0002	0.42 (0.27–0.65)	0.0001
Evaluation				
Pre-intervention	Ref.		Ref.	
Post-intervention	0.47 (0.34–0.66)	<0.0001	0.95 (0.62–1.45)	0.8025

* HH was mandatory per government and school guidance, therefore selected as reference group.

**Table 4 ijerph-22-01198-t004:** Characteristics of hand dirtiness evaluations.

Characteristic	Pre-intervention (N = 82) n (%)	Post-Intervention (N = 126) n (%)
Sex		
Male	41 (50.0)	78 (61.9)
Female	41 (50.0)	48 (38.1)
Age (years)		
5–8	7 (8.5)	29 (23.0)
9–12	71 (86.6)	88 (69.8)
13 or older	4 (4.9)	9 (7.1)
Time		
School entrance	29 (35.4)	42 (33.3)
After restroom	26 (31.7)	42 (33.3)
Before eating	27 (32.9)	42 (33.3)
School location		
Urban	23 (28.1)	42 (33.3)
Rural	59 (72.0)	84 (66.7)

**Table 5 ijerph-22-01198-t005:** Comparison of qPHAT scores pre- and post-intervention.

qPHAT Score	Pre intervention (N = 82) Median (IQR)	Post intervention (N = 126) Median (IQR)	Difference in Median
Overall	6 (3)	8 (3)	2
Time			
At entrance	5 (2)	7 (2)	2
After restroom	7 (2)	9 (2)	2
Before eating	5 (3)	5 (3)	0
Sex			
Male	7 (2)	8 (3)	1
Female	5 (2)	7 (3)	2
School location			
Rural	7 (3)	7 (3)	0
Urban	5 (2)	8 (3)	3

## Data Availability

The data that support the findings of this study are available from the corresponding author, MMP, upon reasonable request.

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
