# Peer review of "A Quasi-Experimental Pre-Post Assessment of Hand Hygiene Practices and Hand Dirtiness Following a School-Based Educational Campaign"

_ijerph, 2025, doi:10.3390/ijerph22081198_

Round 1
Reviewer 1 Report
Comments and Suggestions for Authors
The study evaluated a school-based intervention on observed Hand Hygiene and hand cleanliness in primary school is Guatemala.
The topic is simple but original and meaningful; however, authors are advised to provide sufficient citations from previous studies to justify the gap that their study is seeking to fill. The introduction is well written, but sufficient literature is missing. Authors are advised to include an elaborate literature review comprising previous related studies and their findings to lay a basis for the identification of research gaps. There are several studies on HH, which may not necessarily be exact, but are related.
The study seeks to provide current evidence of HH practices among school children and, in addition, to determine the effectiveness of an implemented HH program in selected primary schools in Guatemala.
The methodology is generally commendably well elaborated.
Specify the age of the target population and sample in the methodology. It is only appearing under the results.
Regarding data from convenient sampling, authors to clearly describe the measures taken to guard against biases associated with the sampling technique.
The conclusion is well written.
References are good, but should be updated with additional references after improvement of the literature review (supporting literature for research gaps).
Similarity is 34.23% it shoud be reduced under 15%.
Author Response
Comments 1: The topic is simple but original and meaningful; however, authors are advised to provide sufficient citations from previous studies to justify the gap that their study is seeking to fill.
Response 1: Thank you for the recognition of the originality and relevance of the topic. In response to your recommendation, we have revised the introduction to include additional citations from previous studies on school-based hand hygiene interventions and have also included national data from Guatemala. These additions help clarify the existing research gap our study seeks to address.
Comments 2: The introduction is well written, but sufficient literature is missing. Authors are advised to include an elaborate literature review comprising previous related studies and their findings to lay a basis for the identification of research gaps. There are several studies on HH, which may not necessarily be exact, but are related.
Response 2: Thank you for your suggestion. We have expanded the introduction to include a more detailed review of related literature on school-based hand hygiene intervention. Specifically, we now reference studies from Kenya and Uganda that demonstrate the impact of educational and infrastructure-based interventions on students’ HH practices, knowledge, and health outcomes. Additionally, we have included data on the current state of HH infrastructure and practices in Guatemala, highlighting the gap in intervention-focused studies in this context.
Comments 3: The study seeks to provide current evidence of HH practices among school children and, in addition, to determine the effectiveness of an implemented HH program in selected primary schools in Guatemala. The methodology is generally commendably well elaborated.
Response 3: Thank you for your positive feedback regarding the study’s objectives and methodology.
Comments 4: Specify the age of the target population and sample in the methodology. It is only appearing under the results.
Response 4: Thank you for this helpful suggestion. We have included the age range of the participants (8 to 13 years old) in the Methods section.
Comments 5: Regarding data from convenient sampling, authors to clearly describe the measures taken to guard against biases associated with the sampling technique.
Response 5: Thank you for your comment. We have addressed the issue of convenience sampling and its associated biases in the limitations section of the manuscript.
Comments 6: The conclusion is well written.
Response 6: Thank you very much for your positive feedback on our conclusion.
Comments 7: References are good, but should be updated with additional references after improvement of the literature review (supporting literature for research gaps).
Response 7: Thank you for your feedback. In response, we have updated the introduction with additional references to strengthen the literature review and better support the identification of the research gap.
Comments 8: Similarity is 34.23% it shoud be reduced under 15%.
Response 8: Thank you for your feedback. We have carefully reviewed the manuscript and used a plagiarism checker (Duplichecker), which indicated a similarity index of 9%. We have taken care to paraphrase existing literature and properly cite all sources to ensure originality.
Reviewer 2 Report
Comments and Suggestions for Authors
Dear Author(s),
Thank you for your effort so far in getting your manuscript to review. The manuscript entitled “A Pre-Post Assessment of Changes in Hand Hygiene Practices and Hand Dirtiness following an Educational Campaign.” This research topic is important. However, these topics have been researched in almost all aspects by many researchers from different regions, both nationally and internationally, for many years, and many useful and striking results have been obtained. I recommend a major revision of this manuscript because it is based on the researchers' direct observations in Guatemala. My suggestions are listed below.
Although the article provides sufficient background information, the methods section is not well planned. The research design should be clearly stated.
The age group of the participants in the study must be included. The manuscript refers to primary school students, but even a one-year difference can cause significant differences in learning outcomes.
It is mentioned that the HH activities were carried out throughout the day, but how many minutes of theoretical and practical information transfer were there for each group? Even if a short film on hand hygiene was shown, this should be detailed.
When were the post-intervention measurements taken? This must be specified.
In comparing pre- and post-intervention measurements, a t-test is used for dependent samples, and this should be clearly stated. Additionally, do the data meet the normality assumption? Depending on this, nonparametric methods may be used in statistical analysis.
Lines 168-170: In the logistic regression analysis, the dependent and independent variables must be specified. Additionally, the “Nagelkerke R Square” value, which indicates the explanatory power of the model, must be provided in Table 3. This means “what percentage of proper hand hygiene practices is explained by the variables given in this table.”
Table 3 is missing the variables before and after the intervention.
In Table 4, it is not appropriate to give a percentage for the qPHAT color scale score. It may be necessary to indicate how many points above the scale are considered problematic. In fact, Table 5 provides the answer to the point I emphasized in my previous sentence. For this reason, Table 4 can be removed or placed in the supplementary material.
The table containing the information in lines 255-261 must be included either in the manuscript or as supplementary material.
It would be nice to see more information in the discussion section about what hand hygiene practices mean for school health policies in Guatemala.
Reviewer 3 Report
Comments and Suggestions for Authors
IJERPH - 3687582 - Research Article – Peer reviewer
Int. J. Environ. Res. Public Health
Title: A Pre-Post Assessment of Changes in Hand Hygiene Practices and Hand Dirtiness following an Educational Campaign
Overview: This research article portrays the results of a pre-post study evaluating school-based intervention on observed hand hygiene practices and hand cleanliness in primary schools in Guatemala, a gap in the literature regarding hand hygiene. Especially relevant after the surge of infectious diseases, this topic is important for addressing the burden of infectious disease, which significantly impact children’s health and educational outcomes.
Recommendation: Reconsider after major revisions
Comments to Authors
Major comments:
- Although the article touches an interesting and relevant topic, the fact that the study was carried out during COVID season implies a significant bias, which clearly impacts the results and therefore cannot be extrapolated to other normal non-pandemic seasons. COVID-19 regulations were also lifted this month and no longer required hand hygiene stations or assistants to be available, which might have impacted willingness to perform said actions, compulsory vs voluntarily, and it does not allow to properly evaluate the intervention studied in this article.
- Another significant source of bias is the selection process of schools to be analyzed, which should be either all-inclusive or randomly selection of schools of different districts or cities, or using some kind of sampling method. Although it is stated in the limitations section and portrayed as a descriptive study, it cannot be extrapolated to the whole of Guatemala, as it implies in the text.
Minor comments:
- In the introduction section, it would be interesting to maybe expand a bit about the background. Although there is a gap in knowledge regarding the topic, it could outline the characteristics of similar interventions or studies in other countries and the impact they had. Besides, said foreign studies could be compared in the discussion section paralleling socio-economical differences and country’s characteristics as to find why it worked or not in some countries or others. Most people in this study field are aware of certain characteristics and effects, but in my opinion the introduction lacks depth.
- In the methods section, the type of study is not specified. Although one can deduce by the description that we are talking about pre-post study with a quasi-experimental design, it should be explicitly outlined or mentioned at the beginning of the methodology section. This is also applicable to the title; being the type of study mentioned in at least one or the other in its entirety.
- The methods section is clear. However, the authors should consider expanding a bit about the statistical analysis as well.
- Following the point #5 above, nowhere in the text is the p-value mentioned. Reevaluate the mention of the p-value in the methods section (or elsewhere for that matter), since the cut-off p-value to be considered significant in this study was not specified in the methods section. Even though one could deduce said values from the tables, it remains quite a bit confusing.
- When analyzing hand dirtiness, the fact that students were selected by convenience instead of using some kind of method to randomize, either even numbers or every x students constitutes another possible source of bias.
- Line 106-107. The fact that schools freely choose how they presented their assigned hand hygiene topics could add an extra possible bias, which could have been prevented by implementing standardized or approved programs to effectively impact hand hygiene.
- Although bias is mentioned in the discussion, it could be expanded, or at least explained how the possible bias was tackled. It could be described if any efforts were made to address potential sources of bias.
- The interpretation of the results and the conclusions drawn from these results are satisfactory, despite the methodological flaws. Nonetheless, in my opinion it would be interesting to delve deeper into future recommendations, suggestions and studies that go beyond just a description.
- Line 394. I would suggest including the DOI of the different papers cited in this article in the references section, for clarity.
Personal Opinion: It is an interesting article since it taps into an important topic that impacts public health and many diseases: hand hygiene. The results highlight the usage trends among a population that is more at risk and more susceptible to learn in the future: children. If the some changes are implemented, its publication could add value to the existing literature on the topic, and aid to shape future programs and policies. However, despite the quasi-experimental design and the characteristics it entails, this paper seems too green and vague on some issues to be a published article.
Round 2
Reviewer 2 Report
Comments and Suggestions for Authors
Dear Author(s),
The authors' response is unclear. I recommend that the authors resubmit their manuscript with the changes they have made indicated by line numbers.
Reviewer 3 Report
Comments and Suggestions for Authors
Accept after minor revisions
Author Response
We thank the reviewer for their comments on our manuscript.
Comments 1: Accept after minor revisions. Study design must be improved.
Response 1: While we recognize the reviewer’s comment regarding the study design, the nature of the project does not allow for changes to the design at this stage. However, we have acknowledged this limitation more clearly in the Discussion section (lines 329-346) and reflected on how it may inform the interpretation of our findings and guide future research. We have also carefully reviewed the manuscript to ensure clarity, correct grammar, and consistency with journal formatting guidelines. We hope these revisions address the concerns raised and look forward to your response.
Round 3
Reviewer 2 Report
Comments and Suggestions for Authors
Dear Author(s),
Thank you for your revisions to the manuscript. The manuscript is ready for publication.